# The Q223R Polymorphism of the Leptin Receptor Gene as a Predictor of Weight Gain in Childhood Obesity and the Identification of Possible Factors Involved

**DOI:** 10.3390/genes11050560

**Published:** 2020-05-17

**Authors:** Helena Marcos-Pasero, Elena Aguilar-Aguilar, Gonzalo Colmenarejo, Ana Ramírez de Molina, Guillermo Reglero, Viviana Loria-Kohen

**Affiliations:** 1Nutrition and Clinical Trials Unit, GENYAL Platform IMDEA-Food Institute, CEI UAM+CSIC, 28049 Madrid, Spain; helena.marcos@imdea.org (H.M.-P.); elena.aguilar@imdea.org (E.A.-A.); 2Biostatistics and Bioinformatics Unit, IMDEA Food Institute, CEI UAM+CSIC, Ctra. De Canto Blanco 8, 28049 Madrid, Spain; gonzalo.colmenarejo@imdea.org; 3Molecular Oncology and Nutritional Genomics of Cancer, IMDEA-Food Institute, CEI UAM+CSIC, 28049 Madrid, Spain; ana.ramirez@imdea.org; 4Production and Development of Foods for Health, IMDEA-Food Institute, CEI UAM+CSIC, 28049 Madrid, Spain; guillermo.reglero@imdea.org; 5Production and Characterization of Novel Foods, CIAL, CEI UAM+CSIC, 28049 Madrid, Spain

**Keywords:** childhood obesity, nutrition precision, *LEPR*, Q223R, weight growth rate, gene-environmental interaction

## Abstract

(1) Background: Childhood rapid weight gain during development has been postulated as a predictor of obesity. The objective of this study was to investigate the effect of single nucleotide polymorphisms (SNPs) on the annual weight gain and height growth, as well as identifying possible lifestyle factors involved. (2) Methods: As part of the GENYAL study, 221 children (6–8 years old) of Madrid (Spain) were enrolled. A total of 11 SNPs associated with high childhood body mass indexes (BMIs) were assessed. Anthropometric measurements, dietary and physical activity data, were collected in 2017 and 2018. Bonferroni-corrected linear models were used to fit the data. (3) Results: A significant association between the Q223R *LEPR* and the weight growth was found, showing a different behavior between GA and GG genotypes (*p* = 0.001). Regarding lifestyle factors, an interaction between Q223R genotypes and total active weekly hours/week to predict the weight growth (kg/year) was observed (*p* = 0.023). In all the genotypes, a beneficial effect against rapid weight growth was observed, but the effect size of the interaction was much more significant in homozygous (GG) minor homozygous (β = −0.61 (−0.95, −0.26) versus heterozygous (AG) and wild-type homozygous (AA) genotypes (β = −0.07 (−0.24, 0.09) and β = −0.12 (−0.32, 0.08), respectively). (4) Conclusions: These results may contribute to more personalized recommendations to prevent childhood obesity.

## 1. Introduction

Pediatric overweight and obesity are the most prevalent nutritional diseases worldwide, developing into a prominent public health problem, both in high- and low-income countries [1]. Recent studies have shown an increasing trend in the global standardized prevalence of this disease. In this context, it is highly indicated to finding variables for the fast detection of risks in children, as well as to find suitable strategies aimed at fighting against obesity [2].

In this respect, rapid weight growth in childhood has been strongly associated with increased overweight and obesity risks later in life [3]. Similarly, it has been observed, an association between metabolic traits such as hypertension in adulthood with a rapid pattern of weight and height growth during childhood [4]. This is the reason why the study of the mechanisms involved in these phenomena can become helpful for obesity prevention strategies.

Moreover, it is well-known that exogenous or common obesity is an acquired pathology associated with lifestyle and high caloric intake [1]. However, conventional treatments up to now, based on a marked nutritional transition as a lifestyle change, have had limited effectiveness [5]. In this sense, the research into the genetics of common obesity has allowed the mapping of single nucleotide polymorphisms (SNPs) to adiposity-related traits. Specifically, in adults, there are up to now more than 100 body mass index (BMI)-associated loci [6]. However, these figures are substantially lower for children, because there are fewer studies performed in this population and the potential impact of confounding factors like age and growth, in addition to genetics or environmental variables [7]. Consequently, the study of the interactions between genes, weight gain, and lifestyle within the new concept of precision nutrition could help to optimize the nutritional counseling programs targeted at people with a particular genotype [8].

On this basis, the objective of this study was to investigate the effect of SNPs associated with high childhood BMI on the annual weight growth and height growth, as well as identifying possible lifestyle factors (dietary and physical activity) involved, in the context of the GENYAL study, to childhood obesity prevention.

## 2. Materials and Methods

### 2.1. Study Design

This study is included in the GENYAL study for childhood obesity prevention, whose primary goal is to develop and validate a predictive model to identify those children who would benefit the most from actions aimed at reducing the risk of obesity and its complications. It is a clinical trial with a 5-year follow-up (2017–2021) based on nutritional education, annual anthropometric measurement evaluations, and data collection from questionnaires about physical activity and dietary aspects.

The education intervention included materials with nutritional information to help parents, students, and teachers to make some lifestyle changes. This program was designed and given by a nutritionist from IMDEA-Food.

Saliva samples were collected for all the schoolchildren in the initial evaluation (2017) in order to obtain genetic information. Given the large number of endpoints and associations examined, it was not possible to perform a rigorous and univocal initial estimation of the sample size. We, therefore, decided to use the largest possible sample according to the available resources. The inclusion criteria to participate in the study were: being in first or second grade of a primary school and having an informed consent signed by at least one of the parents. Exclusion criteria were not attending school during the evaluation days or having planned not to stay at the school in the following years.

First-year follow-up has been used in this study to evaluate the evolution of these variables. Thus, the results shown in this paper correspond to data collected during 2017–2018.

A sample of 221 schoolchildren (116 girls and 105 boys) in first and second grades of primary school (6–8 years of age) from 6 different public schools in the Community of Madrid (Spain) were included at the beginning of the study. The Ministry of Education of this Community was responsible for the sampling of the schools, covering a variety of socioeconomic statuses of the different districts so that the selection was representative of the household income distribution in Madrid defined by the Spanish National Statistics Institute [9].

During the second year of the monitoring (2018), 25 volunteers dropped out of the study (9 children changed schools, 2 had family or medical problems, and 14 of them due to the loss of parents’ interest to participate). Then, a total of 196 children formed the new sample and were analyzed.

### 2.2. Ethical Statement

All parents or legal guardians gave their written informed consent, in which the study management was described. The study protocol was approved by the Research Ethics Committee of the IMDEA-Food Foundation (PI:IM024) and has been registered as a clinical study in ClinicalTrials.gov (NCT03419520). This research follows the guidelines laid down in the Declaration of Helsinki.

### 2.3. Outcome Variables

#### 2.3.1. Anthropometric Measurements

Height was determined using a Leicester height rod with a millimetric accuracy (Biological Medical Technology SL, Barcelona, Spain). Body weight, fat mass (FM) percentage, and muscle mass (MM) percentage were assessed using a body composition monitor (BF511; Omron Healthcare Co., Ltd., Kyoto, Japan). Waist circumferences (WC) were taken using a nonelastic tape (KaWe Kirchner & Wilhelm GmbH, Asperg, Germany; range 0–150 cm, 1 mm of precision). Triceps skinfolds were taken following the International Society for the Advancement of Kinanthropometry guidelines [10] using a mechanic calliper (Holtain Ltd., Crymych, UK; 10-g/mm^2^ constant pressure; range 0–39 mm, and 0.1 mm of precision). Children were assessed at their schools early in the morning by trained nutritionists following standard techniques and the international WHO guidelines specific for this population [11]. Measurements were taken twice in a row, considering the average as the result. To evaluate nutritional status, the percentiles of the International Obesity Task Force (IOTF) were employed [12], and the results of overweight and obesity rates were unified as a single category called excess weight (EW).

Based on these data, the following longitudinal variables were calculated (as the difference between its initial and final value): weight growth: annual change in weight (kg/year) and height growth: annual change in height (cm/year). From all the other anthropometric variables collected (BMI, FM, MM, WC, and triceps skinfolds), the annual change was scaled by the first-year value and, therefore, calculated as the percentage of variation (%V) as follows:(1)$$\%V=\frac{first\;year-second\;year}{first\;year}\;\times 100$$

#### 2.3.2. Dietary and Physical Activity Data

This information was obtained from questionnaires filled out by parents during 2017 and 2018.

Dietary information was gathered using a 48-h food record of two nonconsecutive days: one weekday and one weekend day, as The European Food Safety Authority guidelines recommend [13]. The data were tabulated and analyzed using the DIAL software (Alce Ingeniería, Madrid, Spain) to obtain information about energy intake, macro, and micronutrients [14].

A 48-h physical activity record was collected, corresponding to 24 h of a weekday and an entire weekend day [15]. The time spent doing different activities was multiplied by the corresponding activity coefficient defined by the WHO [16], added, and divided by 24, obtaining the daily coefficient (DC).Then, the DC corresponding to a weekday was multiplied by 5, and the weekend DC by 2, and both results were added and divided by 7, thus obtaining the median individual physical activity coefficient (IPAC) per individual as follows:(2)$$DC=\frac{\left(RAT\;\left(h\right)\;\times \;1\right)+\left(VLAT\;\left(h\right)\;\times \;1.5\right)+\left(LAT\;\left(h\right)\;\times \;2.5\right)+\left(MAT\;\left(h\right)\;\times \;5\right)+\left(IAT\;\left(h\right)\;\times \;7\right)}{24\;h}$$
(3)$$IPAC=\frac{\left(DC\;of\;working\;day\;\times \;5\right)+\left(DC\;of\;weekend\;\times \;2\right)}{7}$$

Rest activities time (RAT), very light activities time (VLAT), light activities time (LAT), moderate activities time (MAT), and intense activities time (IAT).

The total active weekly hours (TAWH) were calculated by considering the time invested in moderate and vigorous extracurricular activities reported by parents and 2 additional curricular hours, which were typically performed at school as physical education and sports classes.

The differences in caloric intake, lipid intake, and TAWH between year 1 and year 2 (Δ) were calculated as the annual change and measured in kJ, g, and hours per week, respectively.

#### 2.3.3. Selection of Single Nucleotide Polymorphisms

For the purpose of this study, 11 SNPs (*BDNF-AS* rs925946, *ETV5* rs7647305, *FTO* rs7190492, *GNPDA2* rs10938397, *KCTD15* rs368794, *LEPR* rs1137101 (Q223R), *MC4R* rs17782313, *NEGR1* rs2568958, *SEC16B* rs10913469, *TCF7L2* rs7903146, and *TMEM18* rs6548238) were selected. These SNPs were included by considering their specific relationship with childhood BMI according previous research, having been identified by genome-wide association studies (GWAS) and the absence of linkage disequilibrium between them.

### 2.4. DNA Extraction and Genotyping

DNA was obtained from saliva samples collected the same day of the anthropometric evaluation. Genomic DNA was extracted according to the protocol described by Stratec^®^ INVISORB^®^ Spin Tissue Mini Kit (INVITEK Molecular GMBH, Berlin, Germany). For genotyping, the DNA samples were loaded in TaqMan^®^ OpenArray^®^ Real-Time PCR plates (Life Technologies Inc., Carlsbad, CA, USA) already configured with the specific selected SNPs with specific waves for each allele marked with a different fluorophore to determine the genotype. This process was made using the OpenArray^®^ AccuFill™ System (Life Technologies Inc., Carlsbad, CA, USA). Once it was ready to be used, a PCR was run, and the chips were read in the QuantStudio^®^ 12K Flex Real-Time PCR instrument (Life Technologies Inc., Carlsbad, CA, USA). The results were analyzed using the TaqMan^®^ Genotyper software (Life Technologies Inc., Carlsbad, CA, USA), which assigns the genotype automatically to each sample according to the amount of detected signal for each fluorophore. Data analysis was made by TaqMan Genotyper Software v1.3 (autocaller confidence level > 90%). Call rates for all SNPs were >96%, and genotype frequencies were in Hardy-Weingberg equilibrium (*p* > 0.05).

### 2.5. Statistical Analysis

A descriptive analysis characterized the sample. Qualitative data were presented as percentages and absolute frequencies, while quantitative data were expressed as mean ± standard deviations. Linear models were used to test the associations between the annual anthropometric changes and the 11 SNPs studied. The genetic variable was included through a co-dominant model with treatment contrasts and using the wild-type genotype as a reference level in order to provide greater flexibility to fit the data. The models were adjusted by sex and age as covariates p-values were corrected by the Bonferroni method for the 11 SNPs.

After finding statistical significance with the *LEPR* rs1137101 (Q223R) SNP, interaction models, adjusted by sex and age, were also considered (annual anthropometric changes, dietary and physical activity variables) for this particular SNP in order to gain some insight on the possible involvement of other variables. The association of weight growth with other environmental variables was also investigated through linear models adjusted by sex and age. For paired proportion comparisons (normal versus excess weight for the three rs1137101 genotypes), a sex and age-adjusted logistic regression was used. All statistical tests were considered bilateral with a significance level of 0.05. Estimated parameters (betas and odds ratios) were obtained with 95% confidence intervals. Statistical analyses were performed using the software R version 3.4 (www.r-project.org).

## 3. Results

The average basal age of participants was 6.75 ± 0.73 years. The global height and weight growths were 5.70 ± 0.99 cm/year and 3.23 ± 1.75 kg/year, respectively. No differences between sexes were found (*p* > 0.05). A significant increase in the percentage of excess weight in children was identified after the second year of evaluation (25.40% to 26.53%; *p* < 0.001), mainly due to the overweight component (16.29% to 18.88%; *p* = 0.008).

No statistically significant associations were found between the 11 SNPs studied and annual anthropometric change variables in the codominant linear models adjusted by sex and age (Appendix A), except between the Q223R *LEPR* and the weight growth and the annual change in waist circumference.

Given these results, we focused on the Q223R *LEPR*. The Q223 genotype frequencies were AA (wild-type homozygous) 31.82%, AG (heterozygous) 50.45%, and GG (homozygous) 17.73%. Table 1 shows the average ± SD change in the total anthropometric variables split by genotype and the β and *p*-values estimated between these variables and the SNP in the codominant model adjusted by sex and age.

The main anthropometric, dietary, and physical activity characteristics according to Q223R genotypes of the children collected during both year 1 and year 2 (2017 and 2018) are summarized in Table 2.

Concerning the basal nutritional status at the first year by Q223R genotype, as shown in Figure 1, the percentage of children with excess weight was greater in the GG than in the AG and AA of this SNP. It shows an increased risk in GG carriers (AA/AG odds ratio (OR) = 0.49 (0.24, 0.99); AA/GG OR = 1.41 (0.62, 3.23); *p* = 0.021.

Linear regression models, adjusted by sex and age, were applied to find out possible factors involved in weight growth. Basal BMI, Δcaloric intake, Δlipid intake, and ΔTAWH were significantly associated with the weight growth variable (β = 0.416 (0.269, 0.563), *p* = 6.33 e-14; β = −0.0008 (−0.002, −0.0002), *p* = 0.016; β = −0.016 (−0.035, −0.003), *p* = 0.012; and β = −0.12 (−0.323, 0.0841), *p* = 0.012, respectively). Nevertheless, when we studied how these association were modulated by the presence of Q223R across the interaction models, only the annual change in the total active weekly hours (ΔTAWH) remained significative (*p* = 0.023). According to this interaction, the GG homozygous genotype reduced, on average, 0.61 kg/year the weight growth for each hour per week of moderate to vigorous physical activity (β = −0.61 (−0.95, −0.26). In the case of the AG and AA genotypes, the slopes were much smaller and not significant (β = −0.07 (−0.24, 0.09) and β = −0.12 (−0.32, 0.08), respectively) (Figure 2).

## 4. Discussion

The GENYAL study found a high prevalence of overweight and obesity, as also having been shown in the literature review at both the regional [17] and national levels [18]. In addition, it was observed that the percentage of children with excess weight increased as they advanced through the school stages. These results further support the need to keep looking for new forms of preventive strategies and targeted treatments in order to increase their effectiveness. [19]. The study of gene-nutrients and gene-environment interactions is one of the proposed strategies framed within the emerging concept of precision nutrition [20].

After the evaluation of the effect of 11 SNPs associated with high childhood BMI on the annual weight and height gain, this study has identified the presence of the *LEPR* Q223R SNP as a weight growth predictor during childhood, and further possible factors involved have been evaluated. Regarding the children’s nutritional status during the first-year evaluation, it could be observed that the GG homozygous carriers for this variant were those that showed the highest percentage of obesity and being overweight.

Leptin, a hormone encoded by the LEP gene, acts as a marker of energy reserves in the hypothalamus through regulating both appetite and energy homeostasis [21,22]. Its regulatory effects are mediated by the binding and activation of the long leptin receptor isoform. The Q223R polymorphism, an arginine (A)-to-glutamine (G) transition at position 223 in the *LEPR* gene, consists of a nonconservative change within the extracellular domain-coding region in the gene. Evidence suggests that the resulting conformational changes of the protein influence the *LEPR* functions and could lead to an abnormal downstream signaling pathway of JAK2 due to a leptin receptor deficiency. The consequences of these abnormalities are leptin resistance and, therefore, an attenuated leptin signaling [23]. Accordingly, it would result in a more significant presence of obesity by Q223R genotype, as we have seen in this study, as well as in previous researches in both pediatric [24,25] and adulthood populations [26,27,28]. However, the functional outcomes for this genetic variant remain poorly defined [29]. The existence of contradictory results [30,31] may be explained either by the gender influence [32] or the underpowered available studies [33].

Rapid weight gain in childhood has been associated with a significant risk for metabolic syndrome later in adulthood, and therefore, it is a critical target for the prevention of overweight and obesity at an early age [34,35]. The current study found a significant association between a greater childhood weight growth and the presence of the *LEPR* Q223R minor allele. This association may indicate a genetic susceptibility to rapid weight growth in homozygotes for the minor allele (GG) in comparison to the heterozygous status (AG). Gallicchio L et al. also found significant differences between AG and GG but only for basal BMI in adults [36]. In this case, we hypothesize that the effects produced by this SNP on weight growth differs from the number of risks alleles and follow a nonadditive pattern. This observation could be related with the sample size and with some cellular genetic interference mechanisms due to a co-dominant effect. To our knowledge, this study is the first reference to the issue, and it could be of great significance in order to assist as a potential predictor of growth and development among school-aged children.

The results presented in this study are in agreement with other researches in which adults carrying the G allele show a more significant weight growth over the years [36,37,38]. In this respect, the identification of the Q223R genetic profile in both adults and children could be of interest to personalized interventions. This result might be explained by the impaired regulation of satiety, produced by the polymorphism due to a decreased leptin action or leptin resistance. A study in Brazilian children identified an association between Q223R and a higher caloric intake, which reinforces this idea [39]. Besides, it has been observed that leptin could serve both as a hormonal signal within the hypothalamic-pituitary-gonadal axis by regulating growth hormone, as well as a skeletal growth factor [40].

Interaction studies were conducted in order to determine possible factors that could influence a faster weight growth in Q223R carriers by considering some of the evaluated dietary and exercise variables. In this way, a significant interaction between the total active weekly hours and the polymorphism, when regressing the weight growth, was identified. This finding suggests that children with minor homozygous genotypes were those who might benefit the most from increasing the time dedicated to moderate and vigorous exercise, thus indicating a possible protective effect against rapid weight growth. The findings observed in this study mirror those of the previous research. For instance, in a nutritional intervention study in prepubescent Polish children, it was observed that, after a lifestyle change program where a physical activity plan targeted to weight loss was included, AA homozygous individuals lost less weight than the GG carriers [41].

Leptin receptors, apart from being located in the central nervous system, are traced in other body tissues, such as skeletal muscle. It has been postulated that this is where leptin regulates glucose and fatty acid metabolism [42]. Moreover, there is experimental evidence supporting that aerobic exercise is effective in modulating leptin sensitivity in skeletal muscles. This fact could reverse the variant effect. Walsh S. et al. observed differences in body compositions after a resistance training intervention depending on the genetic profile; adults with the G allele gained more considerable muscle and fat mass volume than the AA carriers [43]. Such observations could clarify the interaction between SNP and weight growth according to the amount of total active weekly hours. It is, therefore, likely that a physical activity intervention could regulate the pattern of weight growth, mainly in Q223R carriers.

A part of Q223R *LEPR* SNP, there were no statistically significant associations between any of the other 10 SNPs and annual weight gain and/or height growth. There is a lack of studies on this issue based on the literature, except for rs7647305 *ETV5*. Tu W et al. showed a significant association between rs7647305 *ETV5* and quick weight increases in 6 to 17-year-old European-American children through six years of follow-up [44]. The differences in the methodological approach and the age range of the children may explain the disparity in the results.

Finally, the current investigation was limited by the sample size and the use of dietary and physical activity questionnaires, which have been criticized. Nevertheless, in the absence of better, lower cost and high-throughput tools, it can offer valuable information, although it should be interpreted with caution [45]. In addition, the assessment of leptin receptor concentrations in blood would be useful in interpreting the results of this study. However, biochemical parameters were not collected in this study.

While further work is required to confirm the interactions observed in this research, we consider that the identification of this SNP might provide useful input for personalized and individualized early interventions. Notwithstanding, we propose the design and validation in a larger cross-study focused on physical activity intervention as a main prevention and intervention variable. On the other hand, the GENYAL study to childhood obesity prevention is a longitudinal study through primary schooling, which will allow the evaluation of the long-term effects of this polymorphism.

## Figures and Tables

**Figure 1 genes-11-00560-f001:**
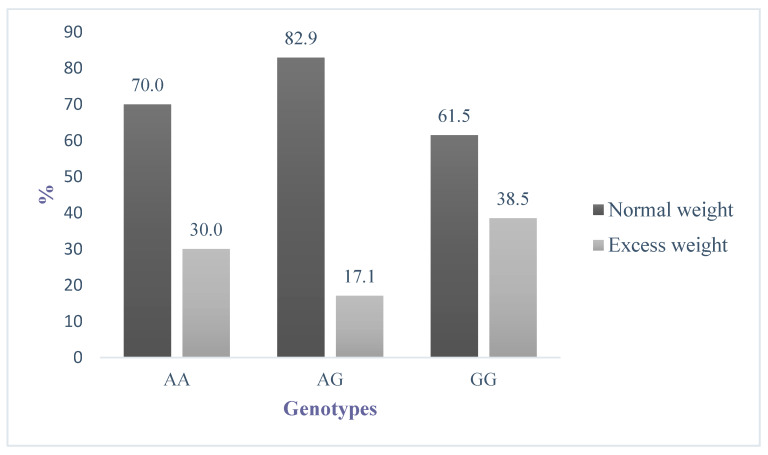
Differences in schoolchildren’s nutritional status according to the wild-type homozygous (AA), heterozygous (AG), and homozygous (GG) genotypes of Q223R. The GG homozygous showed the highest prevalence of overweight plus obesity. The figures presented are taken from basal results (2017). Excess weight comprises obesity and overweight rates.

**Figure 2 genes-11-00560-f002:**
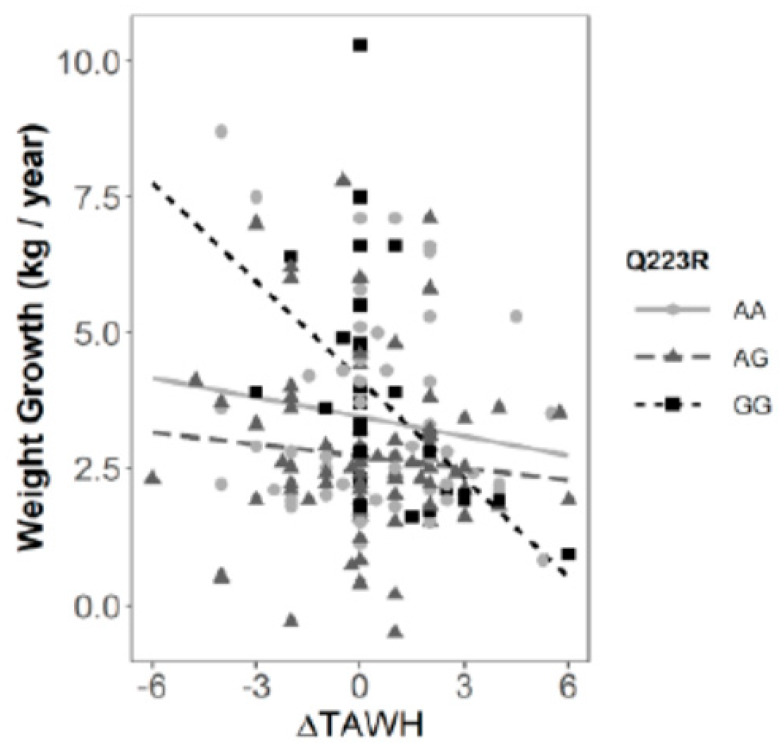
Interaction between Q223R genotypes and ΔTAWH (total active weekly hours/week) to predict weight growth (kg/year). In all the genotypes, a beneficial effect against rapid weight growth was observed with the increase of the time per week of moderate to vigorous physical activity in the schoolchildren, but the effect size of the interaction was much more significant in GG minor homozygous.

**Table 1 genes-11-00560-t001:** Average anthropometric change by genotype, and the association identified between these variables and the presence of the Q223R *LEPR* single nucleotide polymorphism (SNP).

Anthropometric Variables	AA ^1^	AG ^1^	GG ^1^	β (CI) ^2^	*p*
Weight growth (kg/year)	3.52 ± 1.79	2.8 ± 1.49	3.99 ± 2.08	−0.72 (−1.26, −0.18); 0.42 (−0.3, 1.13)	0.001
Height growth (cm/year)	5.77 ± 0.85	5.56 ± 1.06	5.94 ± 1,00	−0.21 (−0.52, 0.1); 0.23 (−0.18, 0.64)	0.063
%V BMI	3.36 ± 5.04	1.63 ± 4.30	3.63 ± 4.87	−1.73 (−3.23, −0.23); 0.24 (−1.76, 2.23)	0.027
%V FM	2.37 ± 13.71	0,00 ± 14.72	1.91 ± 11.76	−2.21 (−6.72, 2.3); −1.09 (−7.06, 4.89)	0.623
%V MM	6.81 ± 4.87	7.74 ± 5.05	6.76 ± 5.04	0.62 (−0.92, 2.15); 0.51 (−1.43, 2.45)	0.723
%V WC	3.8 ± 4.35	1.51 ± 3.91	3.7 ± 4.13	−2.3 (−3.61, −0.99); −0.16 (−1.91, 1.6)	<0.001
%V Triceps fold	2.76 ± 17.55	−1.93 ± 15.68	3.57 ± 12.86	−4.78 (−9.88, 0.33); 1.17 (−5.64, 7.99)	0.075

%V = percentage of variation between year 1 and year 2. FM = fat mass, MM = muscle mass, and WC = waist circumference. ^1^ Mean ± SD. ^2^ Betas (confidence intervals) are provided for both AA/AG and AA/GG. AA (wild-type homozygous), AG (heterozygous), and GG (homozygous).

**Table 2 genes-11-00560-t002:** Anthropometric, dietary, and physical activity characteristics of the study population according to Q223R genotypes in year 1 and 2.

	Q223R *LEPR* YEAR 1 ^1^			Q223R *LEPR* YEAR 2 ^1^		
	AA (WT)	AG	GG	AA (WT)	AG	GG
**Anthropometric variables**						
Weight (kg)	26.61 ± 5.80	25.79 ± 5.48	29.06 ± 7.27	29.55 ± 6.58	28.52 ± 6.33	32.63 ± 8.83
Height (cm)	124.8 ± 6.39	124.08 ± 6.28	126.77 ± 6.40	130.13 ± 6.65	129.74 ± 6.43	132.47 ± 6.32
BMI (kg/m^2^)	16.98 ± 2.79	16.58 ± 2.30	17.85 ± 3.06	17.33 ± 3.00	16.8 ± 2.49	18.32 ± 3.60
Fat mass (%)	20.86 ± 7.50	19.69 ± 6.40	22.79 ± 8.40	21.07 ± 7.89	19.41 ± 6.38	22.46 ± 9.17
Muscle mass (%)	28.07 ± 3.14	27.92 ± 3.05	28.29 ± 2.27	29.57 ± 2.79	29.86 ± 2.45	30.14 ± 1.95
WC (cm)	59.57 ± 6.96	58.88 ± 6.69	62.77 ± 8.79	61.42 ± 8.09	59.47 ± 7.29	64.09 ± 9.79
Triceps fold (mm)	12.75 ± 5.28	11.68 ± 4.60	14.27 ± 5.53	12.89 ± 5.70	11.35 ± 4.97	14.4 ± 6.53
**Dietetic variables**						
Caloric Intake (kJ)	8046.57 ± 1385.79	7634.55 ± 1376.41	7548.55 ± 1502.81	8322.52 ± 1637.54	8110.42 ± 1495.15	7668.04 ± 1670.83
Lipids (g)	85.45 ± 21.09	78.64 ± 18.66	77.27 ± 23.17	86.81 ± 23.27	85.77 ± 22.27	75.5 ± 16.95
SFA (g)	28.71 ± 8.38	27.03 ± 7.57	26.84 ± 8.30	29.1 ± 8.38	28.64 ± 8.50	25.39 ± 6.48
Proteins (g)	78.89 ± 16.05	74.68 ± 13.35	75.59 ± 12.59	88.05 ± 21.33	86.45 ± 17.27	84.64 ± 28.65
Carbohydrates (g)	199.97 ± 34.53	194.82 ± 46.06	193.21 ± 36.53	204.27 ± 45.57	195.61 ± 42.8	195.24 ± 45.87
**Physical activity variables**						
TAWH (hours/week)	4.14 ± 1.96	3.68 ± 1.75	3.33 ± 1.63	4.75 ± 2.35	4.09 ± 2.42	4.3 ± 2.05
IPAC	1.59 ± 0.12	1.59 ± 0.10	1.57 ± 0.11	1.52 ± 0.08	1.53 ± 0.09	1.51 ± 0.08

^1^ Mean ± SD (standard deviations). WC = waist circumference, SFA = saturated fatty acids, TAWH = total active weekly hours, and IPAC= individual physical activity coefficient. WT = wild-type.

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
