# Peer review of "The Q223R Polymorphism of the Leptin Receptor Gene as a Predictor of Weight Gain in Childhood Obesity and the Identification of Possible Factors Involved"

_genes, 2020, doi:10.3390/genes11050560_

Round 1
Reviewer 1 Report
In this manuscript, the authors studied the effects of 11 SNPs on the annual weight and height gains in children, and identified possible lifestyle factors involved. Overall, their findings are interesting and may be of potential clinical use.
There are several concerns regarding to this manuscript.
-In Fig 1, one important information, the y-axis labeling, is missing.
-The LEPR Q223R polymorphism (A to G) could lead to decreased leptin signaling. In this study, the energy intake is similar or relatively lower in GG group compared with AA and AG groups. Is this due to the selected sample size in this study, or other potential reasons, such as diet composition?
-In the interaction study, the author suggested that children in GG group could benefit from the vigorous exercise. Without proper control and detailed study, this seems overreach. Without direct evidence, it is inconclusive if overweight leads to increased annual weight gain, or if higher weight gain results in decreased exercise.
Author Response
We appreciate your comments with regard to submission of our paper entitled "The Q223R Polymorphism of the Leptin Receptor Gene as a predictor of weight gain in childhood obesity and the identification of possible factors involved." (Manuscript ID: genes-790919):
Our comments are in the attached document Response Reviewer 1
Changes are included in the template using the "Track Changes" function.

Reviewer 2 Report
The manuscript by Marcos-Pasero et al. aimed to investigate the effect of 10 SNPs on the annual weight gain and possible lifestyle factors involved in childhood. This clinical study showed a significant association between the Q223R in LEPR and the weight growth and the change in waist circumference. In all the genotypes (AA, AG, and GG) a beneficial effect against rapid weight growth through increased activity was observed but the effect size of the interaction was much more significant in GG minor homozygous genotype. Overall, this is an informative study beneficial for pediatric overweight and obesity research. In general, I supported the publication of this manuscript in the journal of Genes. However, several issues are required to be addressed.
- The major question remains here is why between the AG (heterozygous) and GG (homozygous) genotypes, they display opposite effects on weight gain?
- It seems the fat mass and muscle mass are not significantly changed.
- Did the authors measure hormone levels, such as growth hormone, leptin, etc.?
Author Response
We appreciate your comments with regard to submission of our paper entitled "The Q223R Polymorphism of the Leptin Receptor Gene as a predictor of weight gain in childhood obesity and the identification of possible factors involved." (Manuscript ID: genes-790919):
Our comments are in the attached document Response Reviewer 2
Changes are included in the template using the "Track Changes" function.

Round 2
Reviewer 1 Report
Thanks for addressing the concerns.
Author Response
After consulting with Millie Chen by email (11/05/2020 11:38), she confirmed that we only need to reply to the academic editor comments. That is the reason that we did not add any comments.